# GrASP: Gradient-Based Affordance Selection for Planning

## Abstract

Planning with a learned model is arguably a key component of intelligence. There are several challenges in realizing such a component in large-scale reinforcement learning (RL) problems. One such challenge is dealing effectively with continuous action spaces when using tree-search planning (e.g., it is not feasible to consider every action even at just the root node of the tree). In this paper we present a method for *selecting* affordances useful for planning—for learning which small number of actions/options from a continuous space of actions/options to consider in the tree-expansion process during planning. We consider affordances that are goal-and-state-conditional mappings to actions/options as well as unconditional affordances that simply select actions/options available in all states. Our selection method is gradient based: we compute gradients through the planning procedure to update the parameters of the function that represents affordances. Our empirical work shows that it is feasible to learn to select both primitive-action and option affordances, and that simultaneously learning to select affordances and planning with a learned value-equivalent model can outperform model-free RL.

The ability to plan using a model learned from experience is arguably a key component of intelligence. Repeated planning can help both in the better selection of the action to execute in the current state as well as in providing better targets for updating policies and value functions. A landmark in the success of the use of planning in model-based Deep RL (RL that uses neural-networks as function approximators) was AlphaGo (Silver et al., 2017) in which a simulator of the game of Go was used by the planner as a model. More recently MuZero (Schrittwieser et al., 2020) has shown the capability of learning models from experience simultaneously with planning in solving an impressive number of large-scale RL problems (albeit on problems where a simulator is available and thus sample complexity is not a big concern). Nevertheless, there remain interesting challenges in making planning with learned models become a routine part of Deep RL agents across the spectrum of RL domains. One such challenge, the subject of interest in this paper, is dealing effectively with continuous actions spaces where using tree-search based planning is not straightforward (e.g., just considering every action at the root node of the tree becomes infeasible). Accordingly, we consider RL domains with continuous action or option spaces (where options are temporally extended actions).

Our approach is to *select* a small number of affordances—to discover what actions/options are useful to consider in tree-based planning. The notion of affordances is a rich one in psychology that refers to many kinds of organism-environment relations (Gibson, 1977). We take inspiration in our work from the idea of directly perceived high-level actions that environment states afford, such as object interactions (e.g., chairs afford sitting, cups afford grasping) or navigation actions (doorways afford passage), all restricted subsets of the vast space of possible actions the organism may perform. We implement affordances as parametric mappings from states (and goals in multi-task settings) to actions/options. So selecting $K$ affordances means learning $K$ such mappings, and then there are at most $K$ choices at each node in the planning tree. What action/option the $i^{th}$ affordance corresponds to depends on the state corresponding to a node. We also consider simply selecting $K$ actions/options that are available in all states to evaluate how much conditioning on state matters.

Our main contributions are twofold: (1) the idea of selecting affordances to deal with continuous actions/options in planning, and (2) GrASP an algorithm for **Gr**adient-based **A**ffordance **S**election for **P**lanning. GrASP computes gradients through the planning procedure to update the parameters of the functions that represent affordances. We do not claim or show that GrASP is state of the art for model-based RL in continuous domains. Instead, we show empirically that despite the nonstationarity

of learning to select affordances whilst simultaneously learning a value-equivalent model that predicts their value-consequences, GrASP is fast enough at these two tasks for planning with the discovered affordances and their learned model to often be more sample efficient than model-free approaches in benchmark RL control problems. A comparison with state of the art awaits further development of this new idea of using gradients to discover affordances for planning.

# 1 RELATED WORK

**Other approaches to affordances in RL.** Abel et al. (2014) defines affordances as a subset of actions available to an agent in a state if the state meets certain preconditions defined through propositional functions. They introduce an approach for learning affordances in a scaffolded multi-task learning environment, where affordances learned from simple tasks with smaller state spaces are transferred onto difficult tasks with larger state spaces. With this approach, they demonstrate the utility of affordances to reduce the planning time in a learning agent. Abel et al. (2015) extends this to the setting of goal-based priors: probability distributions over the optimality of each action for a given state and goal pair. Both works are limited in their applicability as they rely on discovering affordances in Object Oriented-MDPs, which assumes objects, object classes, and their attributes as part of the state space (Diuk et al., 2008). It is also limited in the sense that their affordance discovery approach heavily relies on a carefully designed multi-task environment that provides a curriculum to the learning agent. Khetarpal et al. (2020) defined affordances for MDPs and in simple discrete domains showed that hand-specified affordances allow for faster planning in an agent as they reduce the number of actions available at a state and also allow for efficient learning of partial transition models of the environment.

Sherstov & Stone (2005) improved the time required to learn an optimal policy by transferring a reduced action set obtained from the policy of a source task to a new but similar target task. Rosman & Ramamoorthy (2012) provide a method for learning Dirichlet action priors from a set of related tasks that were then shown to improve the planning speed. Other work (Even-Dar et al., 2006; Cruz et al., 2014; 2016; 2018; Zahavy et al., 2018) have also demonstrated the utility of action-pruning to allow an agent to scale up to environments with large action spaces.

In summary, none of the above approaches developed an approach to the discovery of affordances based on gradients through the planning procedure.

**Gradient through a trajectory in a model to update policies.** Another body of work, summarized below, can be seen as effectively using gradients through a trajectory generated from a learned model to update a policy. Our approach to learning affordances using gradients through a planning tree is different in at least two ways. First, we learn affordances that provide a set of actions as a function of state that an agent should consider during planning. Second, we compute gradients through a planning tree rather than a single trajectory. In our empirical work, we consider an ablation of GrASP in which we learn a single affordance mapping, in effect learning a policy via gradients through a trajectory as a means of comparing against an instance of this approach.

PILCO (Deisenroth & Rasmussen, 2011) is an approach that learned state transition models parametrized as Gaussian processes. The policy was derived from gradients obtained by differentiating through a trajectory from the transition model and was shown to learn optimal policies in simple control tasks such as cartpole and mountaincar. Many approaches extended the idea from PILCO (Gal et al., 2016; Higuera et al., 2018; Chua et al., 2018; Amos et al., 2018) to other kinds of models and domains. Dreamer, introduced by Hafner et al. (2020), used a recurrent architecture for learning latent transition models and directly learned the policy parameters from the world model by differentiating the unrolled model with the policy parameters. Byravan et al. (2020) introduced an approach that learns latent deterministic transition models and learned a policy by backpropagating through transition model expansions, similar to that of Dreamer.

**Continuous actions in RL.** Stochastic Value Gradients (SVG) (Heess et al., 2015) allow learning of continuous control policies by obtaining value gradients of one-step model predictions and these value gradients have been shown to have reduced variance, thus producing successful learning algorithms such as Deterministic Policy Gradients (Silver et al., 2014), Deep Deterministic Policy Gradients (Casas, 2017) and Soft Actor-Critic (Haarnoja et al., 2018). These methods use gradients of one-step trajectories to update policies and thus are quite different from GrASP.

**Planning with continuous action spaces.** Progressive widening (Couëtoux et al., 2011) is an early approach for planning in continuous-action domains when a simulator is available. It was recently extended to planning with value-equivalent models by Moerland et al. (2018) and Yang et al. (2020), but both of these have only been shown to work on domains with 1-D actions. MuZero (Schrittwieser et al., 2020) is considered to be a state-of-the-art algorithm for planning with value-equivalent models in RL but was limited to discrete-action domains. More recently an extension of MuZero, called Sampled MuZero (Hubert et al., 2021), was introduced for dealing with continuous actions. Sampled MuZero is different from GrASP in two important respects. Sampled MuZero uses the planning process to update a policy prior distribution over continuous action space and uses samples from this distribution to construct the planning tree, while GrASP uses gradients through the planning process to discover a discrete set of affordance mappings which are used for building the tree. Like in MuZero, our GrASP agents learn value-equivalent models rather than observation prediction models. But unlike MuZero, our GrASP agents also learn option-models in addition to primitive-action models. (We could not directly compare our method against MuZero because no code was released with the paper, and our attempts at a local implementation efficient enough to be usable did not succeed).

## 2 GRASP DETAILS

**Key idea.** Our RL agent is a planning agent that uses an *affordance module* that maps state representations to a small set of $K$ actions or options from a continuous space for use in expanding a look-ahead search tree. The affordance module is represented by a network with parameters $\theta^{\mathrm{afford}}$ and so the planner is, in effect, parameterized by $\theta^{\mathrm{afford}}$. The key idea underlying our method of discovering useful affordances for planning is that we can compute the gradient of performance loss with respect to $\theta^{\mathrm{afford}}$ through the computations of the planner.

The planners we explore here use the affordance module and a learned value-equivalent model to expand a lookahead tree. In principle any tree-expansion procedure could be used including various MCTS algorithms. The key requirement is that performance-loss gradients w.r.to $\theta^{\mathrm{afford}}$ can be computed through the planner's computations, then GrASP can discover affordances online. In our empirical work we will use two tree-expansion procedures: (1) *shallow-depth complete trees* with depth as a parameter. We use learned value functions at the leaf nodes to bootstrap value estimates at the root node and so shallow depth can be sufficient, and the ability of GrASP to discover useful affordances means that we can have very few action choices at each state and thus afford to build complete lookahead trees. (2) *UCT-based MCTS* with number and depth of trajectories as parameters; here we choose affordances from the learned affordances based on an upper-confidence bonus.

**High level overview.** Figure 1 provides an overview of how the planning agent selects actions or options to execute in the environment using the planning tree. The current observation from the environment $\mathbf{x}$ is encoded into an abstract state $\mathbf{s}$ which becomes the root node in the planning tree expanded using $f^{\mathrm{afford}}$ to generate actions/options at each abstract state node that is expanded, and using the value equivalent model to generate next abstract states and predictions of rewards and values. The reward and value predictions are backed-up to produce Q-values for each action or option, which are used to construct a policy over the actions/options at the root node. An action/option is sampled from this policy and executed to produce a transition in the environment. The next observation is encoded to a new abstract-state, from which the agent repeats its planning procedure. There are thus five learned components: the novel affordance module $f^{\mathrm{afford}}$,

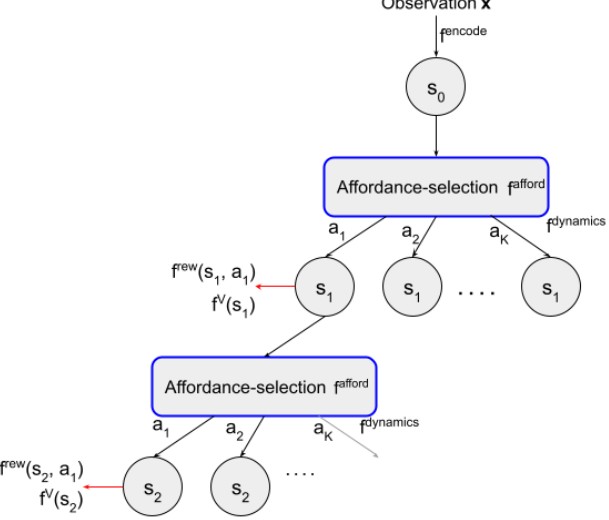

Figure 1: GrASP using its model to perform lookahead planning with actions/options selected via the affordance module. The agent plans by either constructing a $K$-ary depth-$D$ search tree or a partial search tree through UCT.

and the four components of the value equivalent model, an observation encoder $f^{\text{encode}}$, a state-next state dynamics function $f^{\text{dynamic}}$, a reward predictor $f^{\text{rew}}$, and a value function approximator $f^V$. In what follows we specify the key aspects of the algorithm and how the module parameters are learned; full details are in the Appendix.

**Agent-environment interaction.** We assume a discrete time setting in which an agent interacts with a continuous-observation environment using actions or options selected from a continuous space. For the most part we will treat options and actions as if they are equivalent by overloading some terms such as reward and next observation. In the case of selecting actions, at each time step $t$ the agent receives an observation $\mathbf{x}_t \in \mathbb{R}^o$, selects an action $\mathbf{a}_t \in \mathcal{A} = \mathbb{R}^a$ and then receives reward $\mathbf{r}_{t+1} \in \mathbb{R}$ and new observation $\mathbf{x}_{t+1}$. In the case of selecting options, if option $\mathbf{a}_t \in \mathcal{A}$ (note $\mathcal{A}$ is used to denote both the space of actions and options) is selected at time $t$, its policy is followed until termination and the agent receives reward (defined as the discounted sum of rewards from the start of the option until termination), and new observation $\mathbf{x}_{t+n}$ (if the option lasts for $n$ time steps). The next option is then chosen at time $t + n$. The agent's goal is the usual RL goal of maximizing discounted sum of rewards over an episode (all our tasks use a discount factor of $0.99$).

**Affordance module.** The agent encodes each observation $\mathbf{x}_t$ into an abstract state $\mathbf{s}_t$ using an encoding module describe below. The *affordance module* is a network with parameters $\theta^{\text{afford}}$ that maps abstract-states $\mathbf{s}$ to $K$ actions or options $\mathbf{a}^1 \cdots \mathbf{a}^K$ selected from $\mathcal{A}$; i.e., $f^{\text{afford}} : \mathbf{s} \rightarrow \{\mathbf{a}^1, \mathbf{a}^2 \cdots \mathbf{a}^K\}$. In our experiments, we use a simple feedforward neural net architecture for the affordance module with $K$ separate output heads, each head outputting a different action/option.

**Value-equivalent model representation.** The agent plans with a value-equivalent model represented by four neural networks parameterized by $\theta^{\text{M}} = \{\theta^{\text{encode}}, \theta^{\text{dynamic}}, \theta^{\text{rew}}, \theta^V\}$. These four modules are in common with the modules in MuZero's value equivalent model (Schrittwieser et al., 2020); the difference is that our agent does not have a policy-prediction module and that we allow for option-transitions that are temporally extended while MuZero was restricted to action-transitions. The functions computed by each module are defined as follows: The **representation** or **encoding module** $f^{\text{encode}} : \mathbf{x} \rightarrow \mathbf{s}$, parameterized by $\theta^{\text{encode}}$, maps an observation $\mathbf{x}$ to an abstract state $\mathbf{s} \in \mathcal{S} = \mathbb{R}^m$. The abstract state representations are learned; they are not latent environment states or observation predictions. The **value module** $f^V : \mathbf{s} \rightarrow \mathbb{R}$ parameterized by $\theta^V$ estimates the value of an abstract state. The **reward module** $f^{\text{rew}} : \mathbf{s}, \mathbf{a} \rightarrow \hat{r} \in \mathbb{R}$, parameterized by $\theta^{\text{rew}}$, predicts the discounted sum of rewards until termination after executing an action or option $\mathbf{a}$ given state $\mathbf{s}$. For options, the agent learns to predict the expected duration of an option given that it is started in encoded state $\mathbf{s}$. The **dynamics module** $f^{\text{dynamic}} : \mathbf{s}, \mathbf{a} \rightarrow \mathbf{s}'$, parameterized by $\theta^{\text{dynamic}}$ maps an abstract state $\mathbf{s}$ and action/option $\mathbf{a}$ to a predicted next abstract-state $\mathbf{s}' \in \mathcal{S}$.

**Value-equivalent model training.** The parameters of the representation, dynamics and value networks are trained so that the resulting model estimates the value of the environment observation as accurately as possible (hence the term value-equivalent). The reward network is trained to predict the rewards that the agent experiences from a given observation-action/option pair.

Training operates by sampling a sequence of transitions from the replay buffer, as described in pseudocode in Appendix, then using these transitions to update the neural networks of the learning agent. We briefly describe here the learning update of $\theta^{\text{M}}$; the update of $\theta^{\text{afford}}$ is described below. Given a sequence of state-action/option-reward transitions $\{\mathbf{x}_1, \mathbf{a}_1, \mathbf{r}_2, \mathbf{x}_2, \mathbf{a}_2, \mathbf{r}_3, \mathbf{x}_3 \cdots \mathbf{x}_n\}$, the agent first constructs abstract-state $\mathbf{s}_1 = f^{\text{encode}}(\mathbf{x}_1)$, then predicts subsequent abstract states $\mathbf{s}_i = f^{\text{dynamic}}(\mathbf{s}_{i-1}, \mathbf{a}_{i-1})$. The model parameters $\theta^{\text{M}}$ are then updated to minimize the loss:

$$\mathcal{L}^{model} = \sum_{i=1}^{n} \left( \mathbf{r}_i - f^{\text{rew}}(\mathbf{s}_i, \mathbf{a}_i) \right)^2 + \left( \hat{v}_i - f^V(\mathbf{s}_i) \right)^2$$

where, $\hat{v}_i$ is the discounted sum of observed rewards bootstrapped by the predicted value function of the last abstract state in the sequence from a *target* value function network (the value function network is copied into the target value function network periodically). Note that the model loss function is not used to adapt the parameters of the affordance mappings. Model-learning and affordance-discovery are thus independent, though they do constrain each other indirectly because model-learning leads to learning of the abstract state representations which are input to the affordance mappings and similarly the affordance mapping's choice of actions/options produce the trajectories for the replay buffer that are used as data for model-learning.

**Lookahead tree expansion using affordances and value-equivalent model.** In the case of **complete trees**, the planning procedure expands a $K$-ary search tree to depth $D$, expanding each abstract state node with all $K$ actions/options computed by the affordance module for each abstract state. In the case of **UCT**, the planner rolls out $H$ trajectories to depth $D$; the trajectories then implicitly define a partial tree. Each trajectory starts at the root note. If there are one or more affordances that have not yet been selected at the root node, we choose one of those at random. Else, an expansion procedure selects greedily based on the sum of the current Q-values at the root node (see below for backup procedure) and a UCT bonus that boosts the chances of selecting less frequently chosen affordances. At a non-root node whose abstract state has previously been encountered we follow the same expansion procedure (except there is no requirement to select all the affordances at least once). At a non-root node whose abstract state has not been encountered we pick an affordance at random.

**Action selection via planning tree.** Regardless of whether we have a complete tree or a partial tree produced by UCT the following recursive procedure is applied. The value of a leaf-node is the value of the corresponding abstract state from the value network. The value of a non-leaf node is a weighted sum of the Q-values for all the actions/options taken in the partial tree from that node. Suppose the abstract state of the non-leaf node under consideration is $s$, the actions/options in the tree are $a_1, a_2, \cdots, a_f$, the Q-value of $(s, a_i)$ is the sum of the reward from the reward network for $(s, a_i)$ summed with the appropriately-discounted value of the successor node in the tree to $(s, a_i)$. The value of node $s$ then is $\sum_i \pi(a_i|s)Q(s, a_i)$, where $\pi$ is a probability distribution over actions selected in the tree. For complete-tree based planning $\pi(a_i|s) = \frac{exp(Q(s,a_i)/\tau)}{\sum_j exp(Q(s,a_j)/\tau)}$, where $\tau$ is the temperature parameter, while for the UCT-based tree the weight $\pi(a_i|s) = \frac{n(s,a_i)}{\sum_j n(s,a_j)}$, where $n(s, a_i)$ is the count of the number of times action $a_i$ is chosen in state $s$ in the partial tree. Finally, the action executed at the current observation as a result of planning is selected by sampling from the distribution defined by $\pi$ at the root node.

**Gradient update for the affordance module.** The agent updates the affordance module parameters by learning to maximize the value-estimate at the root node $V(\mathbf{s}_0)$. As described above, the value-estimate is obtained via a value backup procedure through the tree, which yields the following expression: $V(\mathbf{s}_0) = \sum_{\mathbf{b} \in \{\mathbf{a}_1, \mathbf{a}_2, \cdots, \mathbf{a}_K\}} \pi(\mathbf{b}|\mathbf{s}_0)Q(\mathbf{s}_0, \mathbf{b})$. where the summation is over the action/option-selections produced by the affordance module. Note that for both the complete tree and the partial tree produced by UCT, each of the affordances is selected at least once at the root. The recursive form of the value function leads to the following recursion to compute the needed gradient:

$$\frac{\partial V(\mathbf{s}_0)}{\partial \theta^{\text{afford}}} = \frac{\partial}{\partial \theta^{\text{afford}}} \Big[ \sum_{\mathbf{b} \in \{\mathbf{a}_1, \mathbf{a}_2, \cdots, \mathbf{a}_K\}} \pi(\mathbf{b}|\mathbf{s}_0)Q(\mathbf{s}_0, \mathbf{b}) \Big]$$

$$= \sum_{\mathbf{b} \in \{\mathbf{a}_1, \mathbf{a}_2, \cdots, \mathbf{a}_K\}} \Big[ \frac{\partial \pi(\mathbf{b}|\mathbf{s}_0)}{\partial \theta^{\text{afford}}} Q(\mathbf{s}_0, \mathbf{b}) + \pi(\mathbf{b}|\mathbf{s}_0) \frac{\partial Q(\mathbf{s}_0, \mathbf{b})}{\partial \theta^{\text{afford}}} \Big]$$

Recall that $\pi(\mathbf{a}_i|\mathbf{s}_0)$ is a function of $\theta^{\text{afford}}$ through the action/option-selections $\mathbf{a}_i$ that are outputs of the affordance module. Finally, $\frac{\partial Q(\mathbf{s}_0, \mathbf{a}_i)}{\partial \theta^{\text{afford}}}$ can be computed via a recursive computation very similar to the value backup procedure:

$$\frac{\partial Q(\mathbf{s}_0, \mathbf{a}_i)}{\partial \theta^{\text{afford}}} = \frac{\partial f^{\text{rew}}(\mathbf{s}_0, \mathbf{a}_i)}{\partial \theta^{\text{afford}}} + \gamma^n \frac{\partial V(\mathbf{s}_1)}{\partial \theta^{\text{afford}}},$$

where $n$ is the expected duration of option $a_i$. This recursive gradient term can be efficiently computed with existing auto-differentiation packages (Griewank & Walther, 2008) with minor additional computational complexity.

## 3 EXPERIMENTS WITH GRASP AGENTS

We report here experiments with GrASP agents on three hierarchical tasks requiring policies over a space of continuous options (note, we assume only the pretrained option policies are available, not the option-model) and on nine domains from the DeepMind Control Suite (Tassa et al., 2018) that requiring policies over continuous primitive-actions. Our first aim is to demonstrate that a planning agent using GrASP is able to learn intuitively sensible affordances and to show that they discover good affordances quickly enough—while simultaneously learning a value equivalent model—to remain

competitive with a strong model-free baseline. Furthermore that GrASP can do so with both the simple complete-tree planner and the more scalable UCT-based planner. We chose TD3 (Dankwa & Zheng, 2019) as our model-free off-policy baseline as it produces stable learning and state-of-the-art performance on many continuous control tasks using a non-distributed agent. Our second aim is to provide evidence that multiple ($K > 1$) affordance mappings are useful for planning, in that the agent's planning-computed policy switches between the $K$ affordance mappings, and yields better performance (asymptotically and in rate) than using any single individual mapping as a policy.

**Neural Network Architecture and Hyperparameters.** The TD3 and GrASP agents all used simple feedforward NN modules. We tuned the learning rate hyperparameters of TD3 and GrASP on the *Point-Mass* task from the DM Control Suite and the tuned hyperparameters were then used across all our experiments. The planning depth for a $K$-ary search tree was tuned on *Fish-Swim* and a depth of 2 was found to work well there. The hyperparameters related to UCT were set to be identical to the ones reported in the MuZero algorithm (Schrittwieser et al., 2020). More details about the NN design and all the hyperparameters used can be found in the Appendix.

### 3.1 LEARNING OPTION-AFFORDANCES IN HIERARCHICAL TASKS

Here we present results on 3 hierarchical control domains in which we provide the agents with pretrained navigation options from a continuous space. In each of the 3 domains, the option space is general but designed to be such that within the space there exist options that are object- or configuration-centric in ways that we would recognize intuitively as sensible environmental affordances to use when planning. Furthermore, the space of multi-tasks for each domain is designed such that for GrASP to perform well across tasks it ought to find the intended affordances for planning. The main evaluation goal is to see if GrASP succeeds in doing so.

**Tasks and option spaces:**
*Collect for Object-Centric Affordances* We designed our first domain and tasks so it would be easy to visualize the affordances, and where to achieve high performance the affordances should be object-centric, i.e., options that are in some way oriented toward objects in the environment. The environment is a continuous 2D world where at the start of each episode the agent and three different objects, denoted A, B, and C, are placed in random positions. A task, encoded in a goal vector given to the agent, involves collecting the three objects in a particular order. The agent receives a reward of 1 when it collects the objects in the correct sequence. The episode terminates without any reward when the agent picks an object out of order. There are five discrete primitive actions: *up,down,left,right* actions which move the agent in discrete steps of size $\epsilon$, and a *collect* action collects an object only if it is within $\epsilon$ distance of the agent. We defined and pretrained a 2-D continuous space of *navigate-and-collect* options that move the agent to within $\epsilon$ of an $x, y$ position and then execute the *collect* action. Note that the option space is not object-centric, but we expect GrASP to learn an affordance mapping that selects the 3 options corresponding to collecting the 3 objects.

*Ant-Gather for Object-Centric Affordances.* The second domain and tasks involves navigation with an Ant agent. The agent starts at a random location with 3 apples and 5 bombs that are placed at fixed locations. The task is to navigate to and collect the apples while avoiding the bombs. The agent receives a reward of 1 when it collects an apple and 0 when it collects a bomb. We pretrained a space of options to achieve any feasible $x, y$ locations with $u$ and $v$ velocities; so the space of options is not object-centric. Again, the goal is to see if GrASP can discover object-centric affordances this time with the additional aspect of avoiding certain objects.

*Point-Mass for Configuration-Centric Affordances.* The third domain and tasks is adapted from the Point-Mass domain in DM Control Suite. The agent is a point-mass and must navigate and pass through three fixed locations in a specific order that is encoded in a goal-vector given to the agent. The agent receives a reward of 1 at the episode termination if it crossed the three locations in the correct order, otherwise the reward is 0. We pretrained a 4-D option space to achieve any feasible $x$-$y$ location with any $u$ and $v$ velocity. In this case, the intended affordances are specific goals of achievement in configuration space (location and velocities).

**Variants of the GrASP -based planning agents.** We create distinct GrASP agents by varying type of *affordance mapping*, *number of distinct affordance mapping heads*, and *planning algorithm* for the complete-tree lookahead at depth $D = 2$. We explored two kinds of affordance mappings: mappings conditioned on both abstract state and goal configuration, which we refer to as *Goal-conditioned Affordances* (abbreviated GA), and mappings that do not condition on either states or goals, which

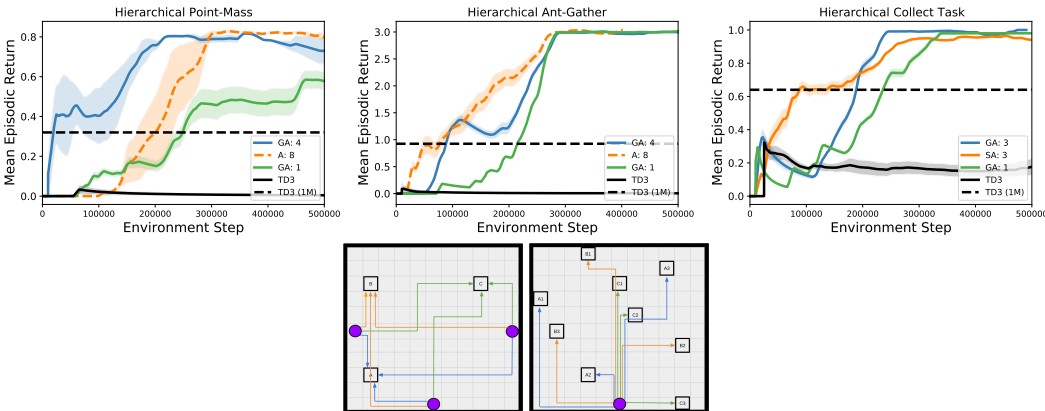

Figure 2: *Top:* Learning performance of GrASP agents and the TD3 baseline on 3 hierarchical tasks with pretrained continuous options. *Bottom:* Visualizations of object-centric options discovered by the GrASP SA-3 agent in the *Collect* task. Left panel shows the trajectories generated by the option chosen by each of 3 affordance heads, in 3 different colors, from three different starting states for the agent. Right panel shows the same for a single start state but varying positions of objects.

we refer to as *Actions* (abbreviated A), because they are similar to conventional agent action sets that do not vary by state. For each kind of mapping we also explored varying the number of distinct mappings $K$, considering $K = 1, 2, 4, 8$ for Goal-conditioned Affordances (abbreviated GA-1, GA-2, GA-4, GA-8), and $K = 4, 8$ for Actions (abbreviated A-4, A-8). We did not explore A-1 or A-2 because it is not possible to solve these tasks with one or two fixed actions in the continuous space. Note that GA-1 has a special status because it is collapses the search tree to a single trajectory and so implements in our framework the main idea in the algorithms described in the Related Work section *Gradient through a trajectory in a model to update policies*. In the *Collect* domain we also tested a GrASP agent, SA-3, that has 3 affordance heads that condition on *state* but not on the goal vector.

**Results.** Figure 2 (top) shows the learning performance for the TD3 baseline, GA-1 (uses the same option-space as the GrASP agents), and the best performing conditioned Affordances and Action agents, GA-4 and A-8 (See Appendix for all learning curves). Each learning curve represents a mean over 5 random seeds and the shaded regions denote their standard errors. We note the following: (1) The GrASP agents (GA-4 and A-8) learn much faster and achieve much higher levels of performance than TD3 in all three tasks. In 500K steps the TD3 agent shows almost no learning; we continued training TD3 until 1M steps and plots its performance at that point as the dashed horizontal line. (2) In two out of three tasks, GA-4 outperforms A-8. (3) In all three tasks, GA-4 learns significantly faster that GA-1, demonstrating the benefits of tree-based planning with option-affordances. Figure 2 (bottom) visualizes the option affordances selected by the GrASP SA-3 agent in *Collect*; it is clear the agent has learned that the best options to plan with are those that go to objects. In effect it has learned object-centric affordances in this environment. Similarly, the affordances learned in *Ant-Gather* are object-centric in that they navigate to apples (while avoiding bombs) and the affordances learned in *Point-Mass* correspond to goals of achievement in configuration space.

## 3.2 LEARNING ACTION-AFFORDANCES IN DEEPMIND CONTROL SUITE

We evaluated GrASP on the following DM Control Suite tasks (Tassa et al., 2018): *Reacher-Easy, Cartpole-Swingup, Point-Mass-Easy, Ball-In-Cup-Catch, Fish-Swim, Fish-Upright, Cheetah-Run, Finger-Spin* and *Walker-Walk*. We selected these because a standard model-free RL agent can learn a close-to-optimal policy within 1M environment steps (adopting Srinivas et al. (2020)'s criterion). All agents observed the low-level state based features, not pixel observations. Agents also observed a target *goal configuration* which varied from episode to episode. In the appendix, we present results showing that GrASP is comparable to Dreamer (Hafner et al., 2019) (whose code was available) in the experimental setting used by Dreamer in which the agent observes pixels. In addition to the GrASP -variants introduced in Section. 3.1, here we also consider versions using UCT for which we did a more limited exploration of the space of possible parameters: we fixed the number of affordance mappings to 4, used only goal-condition affordances, and varied number of trajectories to 20 and 50 (denoted UCT-20 and UCT-50 respectively).

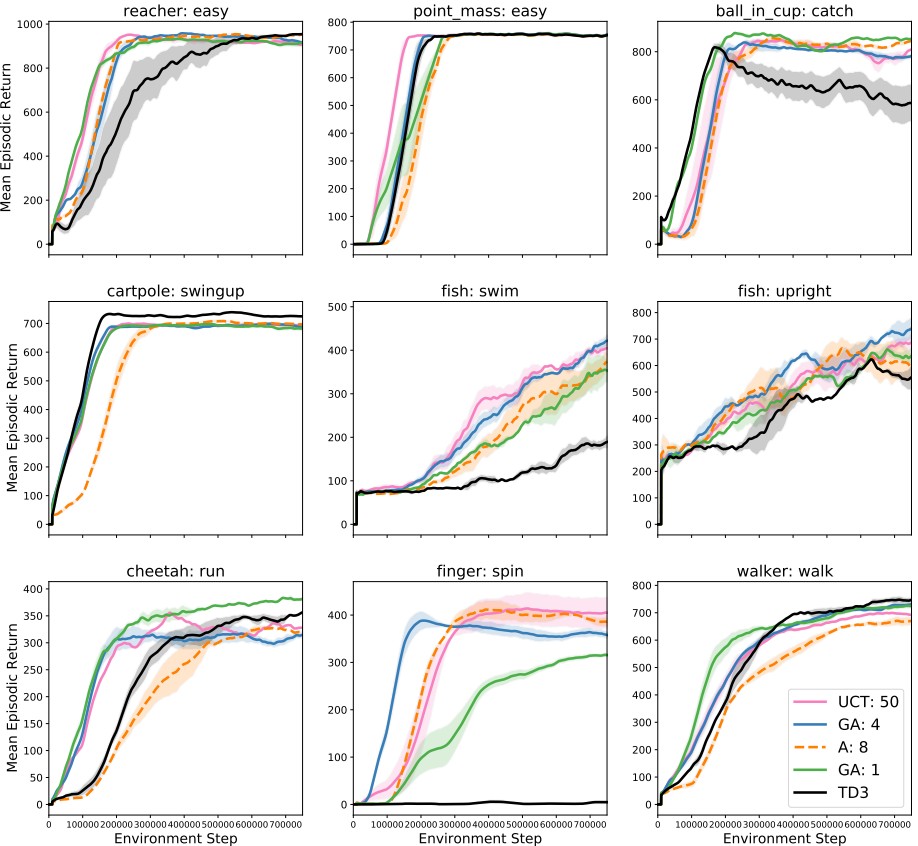

Figure 3: Learning performance of four GrASP agents and the model-free baseline TD3 on nine DM Control Suite tasks. UCT-50 is the best performing UCT-based GrASP agent; GA-4 is the best performing of the Goal-conditioned Affordance agents; A-8 is the best performing of the unconditioned-Actions agents; GA-1 is of special interest because it collapses the tree to a single trajectory; see text. Learning curves for all agents are in the Appendix.

**Results.** Figure 3 shows the learning performance for the baseline TD3, GA-1, the best performing agent of the Goal-conditioned Affordance and Action agents, GA-4 and A-8, the best performing UCT-based GrASP agent UCT-50. (We show this subset to make reading the graphs easier. See Appendix for all learning curves; most GrASP variants outperform TD3 in most domains). Each curve is a mean over 5 random seeds; shaded regions are standard errors. Note the following: (1) All three of the good GrASP agents (GA-4, UCT-50, and A-8) learn faster or achieve better asymptotic performance than the TD3 baseline in 8 out of the 9 tasks. *Cartpole-Swingup* is the only domain where all configurations of GrASP fail to improve over the TD3 baseline agent. (2) In most tasks (8 out of 9) the GA-4 GrASP agent learning conditioned affordances performs better than the A-8 GrASP agent learning to select unconditioned-actions. *Finger-Spin* is the only task where GrASP learning actions produces better performance than agents learning affordances. (3) The GA-4 agent learning multiple ($K = 4$) affordance mappings produces better asymptotic performance than GA-1, which is learning only one affordance mapping, in 4 of the 9 tasks. In 3 of the 9 tasks, the GA-1 agent is better, and in the other 2 tasks there is little difference. (4) The UCT-50 agent is not a clear winner over the complete-tree based agents, doing slightly better in a couple of domains but slightly worse in the others; this may be because discovering affordances while bootstrapping with learned value functions allows for the simpler shallow-depth complete-tree planning to compete well with the more sophisticated UCT-based planning.

**Analysis of switching between affordance mappings.** A single conditioned affordance mapping has the *form* of a policy mapping states/goals to actions/options. The outputs of multiple affordance mappings are interesting *as affordances* to the extent that the agent's planning-computed policy *switches* between the affordance mapping outputs, and the switching policy performance is better than using any single affordance mapping as a policy.

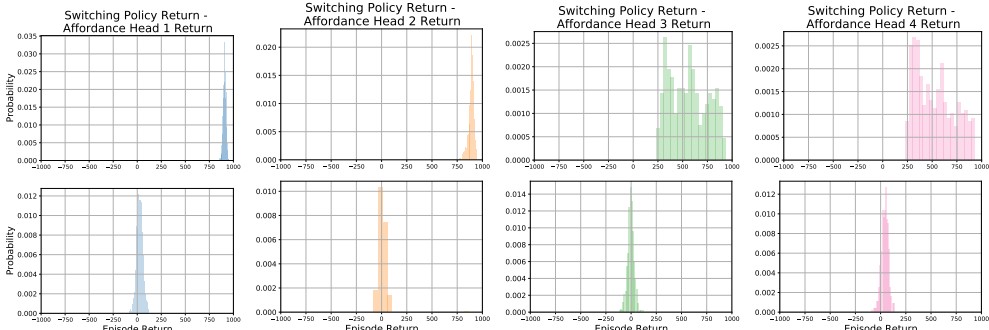

Figure 5: *Top*: Finger-Spin; *Bottom:* Walker-Walk. Distributions of differences in returns between the GA-4 agent planning with discovered affordances and agents following the policies of single affordance-mapping heads.

We conducted two qualitative analyses of switching. In the first analysis we confirm that the agent policies do switch between affordance mappings by producing simple visualizations of the switching behavior of the trained policies across the domains. In all domains we observe rapid switching for all GA-k agents $k > 1$. Figure 4 shows an example in *Fish-Swim* of how GA-4 switches between actions computed by the 4 distinct affordance heads. In the $2^{nd}$ analysis we compare the returns of the switching policy obtained by planning over the trained affordances to the returns of agents that use a single one of the $K$ affordance mappings as a policy.

For a 1000 distinct start-goal configurations we computed performance difference scores between the planning-with-affordances agents and the affordance-mapping-as-policy agents. We summarized these differences with histograms for each affordance-mapping head. If these distributions are skewed to the positive side of zero, it implies that planning with discovered affordances obtains returns that are generally higher than following a single affordance-mapping policy. Figure 5 shows two interesting and representative cases (full results in the Appendix). *Finger-Spin* is a domain where GA-4 showed substantial gains over TD-3 and GA-1, and the distributions of difference scores is positively skewed for all 4 affordance heads. For *Walker-Walk* the asymptotic performance of GA-4, GA-1, TD-3 are comparable, and the distributions of differences are tightly clustered around zero. In general, across the domains and across the GA-k $k > 1$ agents we see positively skewed distributions when those agents outperform GA-1 and TD-3.

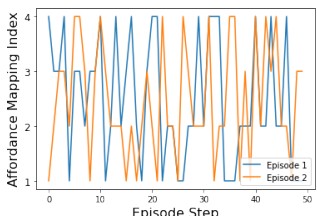

Figure 4: A visualization of the GA-4 agent switching between actions output by the $K = 4$ distinct affordance heads. Partial episodes are from *Fish-Swim*.

## 4    CONCLUSION AND FUTURE WORK

The two main contributions of this work are (1) the idea of discovering affordances—actions or options that are considered at each imagined state in planning—to address the challenge of planning with continuous actions/option; and (2) GrASP , a gradient-based method for discovering good action and option affordances by computing gradients through planning procedures to update the parameters of functions that compute the affordances. We demonstrated the ability of GrASP to learn both primitive-action and option affordances quickly enough while simultaneously learning a value-equivalent model to yield performance on multiple continous control tasks that is competitive with—and often better than—a strong model-free baseline. Analysis of the agent policies shows clear evidence that the learned affordance mappings are useful as affordances for planning and not merely as policies. We provided evidence for the generality of GrASP by combining it with two distinct planning procedures—a simple complete-tree lookahead planner and the more scalable UCT—and by showing its effectiveness in discovering both primitive-action and option affordances. In our view, the main limitation of the work presented here is that we have not yet integrated our affordance discovery into a sophisticated model-based RL algorithm like MuZero that employs a number of other ideas to scale to large domains. This is a clear and exciting next step.

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

# A APPENDIX

## A.1 HARDWARE USED FOR OUR EXPERIMENTS

Each agent was trained on a single NVIDIA GeForce RTX 2080 Ti in all of our experiments.

## A.2 EXPERIMENTS TO COMPARE WITH A MODEL-BASED RL APPROACH

In this section, we compare against Dreamer (Hafner et al., 2019) which is a model-based approach that learns from pixel observations. Their experimental setup is different from those of ours considered in Section 3.2 in that they use pixel observations and they also use an action-repeat of 2. So we ran additional experiments with GrASP (GA=4) to match the experimental setting of that of Dreamer and compare their results here. We used Dreamer's open-sourced code for producing their learning curves. The results are averaged from 3 random seeds.

Overall GrASP performs comparably with Dreamer matching or surpassing its performance in 3 out of 4 tasks. GrASP in *Finger-Spin* reaches a higher asymptote and in *Ball-In-Cup-Catch* it seems to be learning relatively faster than Dreamer. The asymptotic performance is comparable in *Ball-In-Cup-Catch* and *Walker-Walk*, while Dreamer learns better in *Cheetah-Run*.

These results show that GrASP can handle tasks with pixel observation and can produce comparable performance with a state-of-the-art model-based RL approach. We believe that this comparison along with results from Sections 3.2, 3.1 and the visualizations on hierarchical RL tasks clarifies our contribution, which is a method that identifies affordances that are useful for a planning agent (rather than improving state-of-the-art performance in specific continuous control tasks).

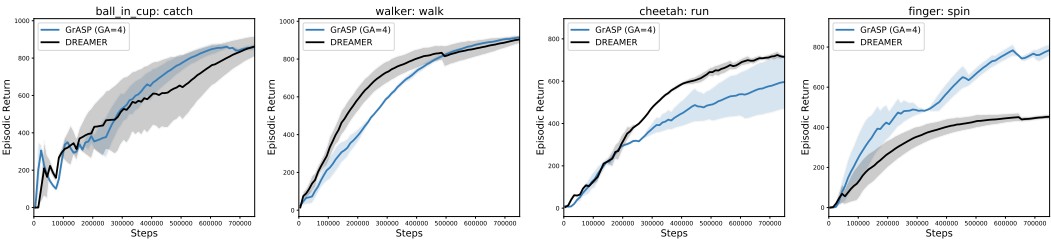

Figure 6: Learning performance of GrASP and Dreamer agents on four DM Control Suite tasks. We present GA=4 version as it was one of the best performing GrASP agents in Section. 3.2.

## A.3 SWITCHING BETWEEN AFFORDANCE MAPPINGS

Figure 7 shows the distribution of average episodic returns from 1000 episodes. Here we compare the returns of the switching policy obtained by planning over the trained affordances to the returns of agents that use a single one of the K affordance mappings as a policy. We summarized these differences with histograms for each affordance-mapping head. If these distributions are skewed to the positive side of zero, it implies that planning with discovered affordances obtains returns that are generally higher than following a single affordance-mapping policy. We conjecture that if all the outputs of the affordance module collapsed to the same output, then this figure would be a lot different: we would have seen the distributions to have concentrated around 0. However, this is not the case in many of the domains: In Finger-Spin (shown in main text; Figure 5 top row), Fish-Swim, Cheetah-Run, Fish-Upright and Reacher-Easy, the distributions are not concentrated at 0 and are spread over the range implying that these affordances are indeed different from each other. In Walker-Walk (shown in main text; Figure 5 bottom row) and Ball-in-Cup-Catch, the distributions are concentrated at 0, which hints at the affordances collapsing to similar outputs.

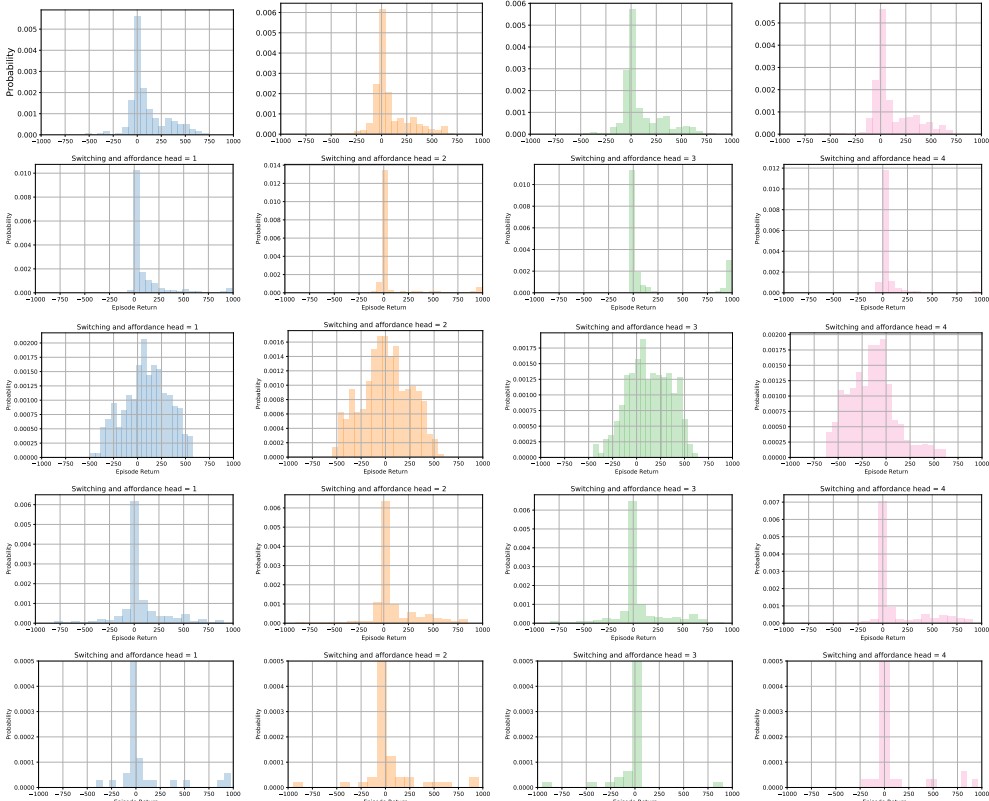

Figure 7: *First Row*: Fish-Swim; *Second*: Ball-in-Cup-Catch; *Third*: Cheetah-Run; *Fourth*: Fish-Upright; *Fifth*: Reacher-Easy. Each row corresponds to a DMControl Suite Task and summarizes the distributions of differences in returns between the GA-4 agent planning with discovered affordances and agents following the policies of single affordance-mapping heads.

## A.4    ADDITIONAL VISUALIZATION FROM COLLECT DOMAIN

The visualizations in Figure 2 of the main text is from the SA-3 agent where the affordance heads condition on the state but not on the goal vector. For the GA-3 agent, the affordance heads condition on the state and goal vector, and so can essentially just learn affordances that directs the agent in picking up the next object that is required of the task. It has no incentive to learn to associate each affordance with collecting each of the objects at each state of the environment as it has access to the goal information. The SA-3 agent on the contrary does not have the goal information and has to learn affordances that can allow it to solve all possible tasks defined in this environment.

The visualization of the affordances learned by the GA-3 agent is shown in Figure 8, where the agent's start position is varied across the domain and trajectories from each of its affordances are shown. The task requires the agent to first collect the B object and one of the affordances does associate with collecting that object. The remaining affordances however are leading the agent to different parts of the environment.

This visualization from GA-3 again shows that the affordances are learning something different from each other and are not collapsing to the same output.

## A.5    COMPLETE LEARNING CURVES

We explored distinct GrASP agents by varying type of affordance mapping, number of distinct affordance mapping heads and planning algorithm for the DM Control tasks. We explored two kinds of affordance mappings: Goal-conditioned Affordances (GA) and mappings that do not condition on either states or goals, which we refer to as Actions (A), as they are similar to conventional action

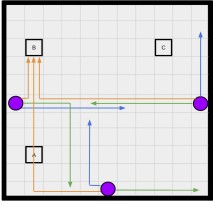

Figure 8: Visualizations of object-centric options discovered by GrASP GA-3 in the *Collect* task. The figure shows the trajectories generated by the option chosen by each of 3 affordance heads, in 3 different colors, from three different starting states for the agent. The task is fixed for each of the three starting states of the agent.

space of an RL agent that are available in all states within an environment. For the GA mapping, we considered different number of affordance mapping heads $K = 1, 2, 4, 8$ (correspondingly abbreviated as GA-1, GA-2, GA-4 and GA-8), and $K = 4, 8$ for Actions (abbreviated as A-4 and A-8). For the experiments studying different number of affordance mapping heads, we fixed the planning procedure as a complete-tree lookahead of depth $D = 2$. We also explored using GrASP agents with UCT as the planning procedure, instead of the complete-tree lookahead approach, for which we did a limited exploration of the space of possible parameters. We fixed the type of affordance mapping to Goal-conditioned Affordances and the number of affordance mappings heads as 4, the number of simulation trajectories for UCT was varied to between 20 and 50.

In the main text, for the DM Control results, we presented learning curves from GA-4, A-8 and UCT-50 as they uniformly produced better learning performance when compared to similar variants with different parameter settings. In Figure 9, we present the learning curves for different number of affordance mapping heads that we considered for Goal-conditioned Affordances and for Actions. We also present learning curves obtained from the GrASP agents using UCT in the same figure.

Identical experiments to the ones described for DM Control tasks were conducted for the Hierarchical Point-Mass and Ant-Gather tasks, and complete learning curves from those experiments are presented in Figure 10. Since the UCT-based agents performed on similar levels to that of complete-tree lookahead agents on DM Control tasks, we did not explore using UCT on the hierarchical tasks.

### A.6 IMPLEMENTATION DETAILS

**Neural network architecture.** We used the following NN architecture for the GrASP agent. The same architecture was used in both DM Control and Hierarchical tasks. We used a three-layer MLP for the representation module. Each layer used 512 units. The reward, value and dynamics modules used a two-layer MLP, each with 512 units, to map the abstract state input into their respective outputs. The dynamics module maps from an abstract state and action/option selection to the next abstract state. The input to the dynamics module consists of concatenation of abstract state and action/option selection that is first embedded using two-layer MLP, again with 512 units each. The affordance module also used a two-layer MLP with 512 units each to map an abstract state to the multiple output heads, where each output head corresponds to an action/option selection made by the agent. The output of the affordance module was bounded by applying $tanh$ activation. ELU activations were used throughout for all MLP layers.

The TD3 baseline agent consists of separate actor and critic modules, and as per its original presentation (Heess et al., 2015), no parameters were shared between those modules. The actor and critic used three-layer MLP each with 512 units which transformed their inputs to their respective outputs. ReLU activations were used for the MLP. The input to the actor is simply the observation received from the environment, while the input to the critic is the concatenation of the observation and the action selected by the agent at the same time step. The output of the actor is bounded between $-1$ and 1, which is achieved by applying $tanh$ activation. This architecture is similar to the one used for the DDPG agent in Tassa et al. (2018).

**Hyperparameters used in our experiments.** We tuned the learning rates for the TD3 agent and the GrASP agent with Goal-conditioned Affordances with $K = 4$ affordance mapping heads on the *Point-Mass* task from the DM Control Suite. We used this tuned learning rate for the rest of the

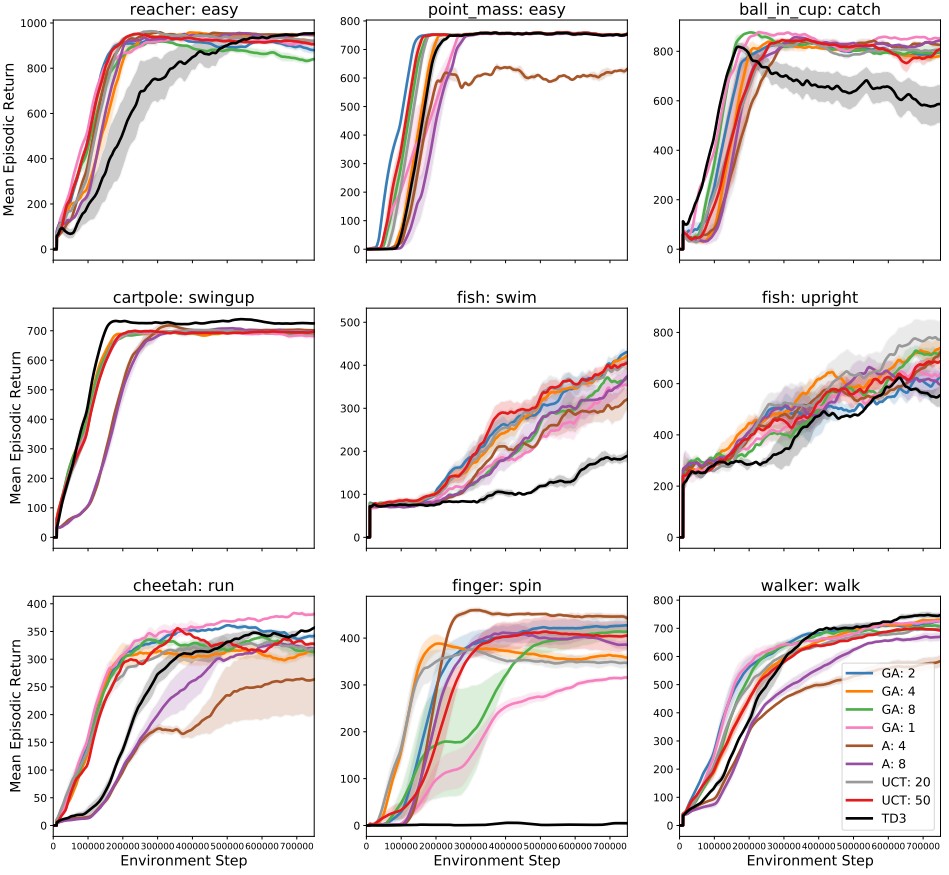

Figure 9: Learning performance of different variants of GrASP agent and the model-free baseline TD3 on nine DM Control Suite tasks. Recall that GA-1 is of special interest because it collapses the tree to a single trajectory. We presented a subset of the learning curves from this Figure as the main result in our main text.

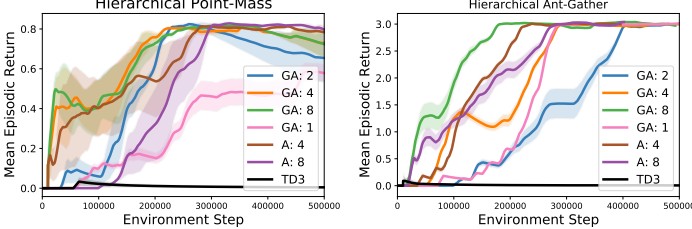

Figure 10: Learning performance of different variants of GrASP agents and the model-free TD3 baseline on 2 hierarchical tasks with pretrained continuous options. All the agents shown here learn to select from the same pretrained space of option policies, thus making a fair comparison over the hierarchical agents. A subset of the learning curves from this Figure was presented in our main text.

DM Control Suite and for the Hierarchical tasks. We also used the same learning rate for different variants of the GrASP agent. The range of values for the initial learning rate hyperparameter was: $\{0.0001, 0.0003, 0.0005, 0.0007, 0.001, 0.003\}$. For TD3, we tuned the learning rates of the actor and critic independently and found that an actor learning rate of $0.0001$ and a critic learning rate of $0.001$ produced optimal learning on *Point-Mass* task. We tuned the learning rate for the affordance module and found $0.001$ to produce optimal learning on the task. We did not tune the learning rate for the representation, reward, value and dynamics modules and set them arbitrarily to be $0.0001$. We used an ADAM optimizer for learning the parameters of all the modules in TD3 and GrASP . The

hyperparameter $\epsilon$ for the optimizer was set to its default value which is $1 \times 10^{-8}$. The action-repeats for the DM Control Suite were set to their default values and were not modified. The discount factor $\gamma$ was set to be 0.99. A replay buffer capable of storing 200k transitions was used for both GrASP and TD3 agents; the samples were drawn uniformly from this replay buffer. The TD3 baseline used soft updates to synchronize the target network with the online network. A smoothing factor of $\tau = 0.005$ was used for these soft updates. The GrASP agents used hard updates to synchronize the target network parameters with the online network parameters and synchronized every $T = 1000$ learning updates. We experimented using soft updates for GrASP agents but found that using hard updates produced better improvements than the version with soft updates.

**Hierarchical tasks.** In addition to the description provided in the main text, we provide more details about the hierarchical domains here.

*Collect:* This hierarchical task is as described in the main text.

*Point-Mass:* The agent's start position is randomly initialized from all possible states in its environment. The task for the agent is to pass through the three objects at fixed locations in a specified order (defined through a goal-vector) to receive a reward of 1. The episode successfully terminates when the agent receives a reward of 1 (i.e., successfully pass over the three objects in the required order). If the agent fails to complete the task, then the agent receives a reward of 0 and the episode terminates when 500 steps are reached. As there are three objects, there are 6 possible ways of ordering the three objects. Thus, there are 6 tasks defined using this environment.

*Ant-Gather:* This task is adapted from the Ant-Gather task introduced in Florensa et al. (2017). The task used in our experiments consists of three apples and five bombs located at fixed locations. The agent's start position is randomly initialized from the possible states from this environment. The episode terminates when the agent either successfully navigates and collects all three apples from the environment or when the number of episode steps reaches 500. The agent receives 1 as reward when it collects an apple and 0 when it collects a bomb.

# B  ADDITIONAL EXPERIMENTAL RESULTS

In this section, we present additional experiments that were designed for evaluating some of the design choices involving the GrASP agent. The design decisions made from these experiments were subsequently used for obtaining the results with the GrASP agent that were presented in the main text.

**Planning depths.** Figure 11 *Left* shows the learning curve from three GrASP agents on *Fish-Swim* task from DM Control Suite, where all three agents use a complete-tree lookahead search but plans with depths $D = 1, 2, 3$ respectively. The GrASP agent used Goal-conditioned Affordances with $K = 4$ affordance mapping heads (i.e., GA-4). From this experiment, we observed a clear ordering of the agent's rate of learning w.r.to the planning depth: GrASP with $D = 1$ learned significantly faster than GrASP with $D = 2, 3$. However, based on the final performance, GrASP with $D = 2$ marginally outperformed the other agents with $D = 1$ and 3 planning depths. Thus, we selected to use $D = 2$ in all our main experiments.

**GrASP agent that uses randomly initialized goal-conditioned affordances and actions.** Figure 11 *Right* shows the learning performance of different agents on the *Fish-Swim* task. Specifically, the plot shows learning curves of GrASP agents GA-4 and A-4, and TD3 baseline. The plot also shows the performance of the GrASP agents that uses randomly initialized goal-conditioned affordances (RND GA-4) and actions (RND A-8). In other words, the RND GA-4 and RND A-8 agents do not learn the parameters of the affordance module. From this figure, we can clearly see that it is essential for the GrASP agent to learn the parameters of its affordance module in order to produce any learning on a given task. As these RND GA-4 and RND A-4 agents failed to produce any learning on the given task, we do not explore those agents on the remaining DM Control and Hierarchical tasks.

## B.1  PSEUDOCODE FOR GRASP

Algorithm 1 presents the pseudocode for the GrASP agent that learns to select affordances in the form of actions or options.

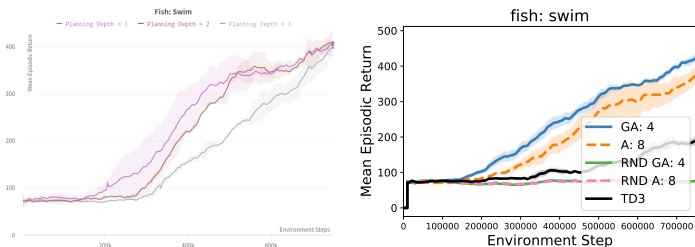

Figure 11: *Left:* Learning performance of GrASP agents with different planning depths $D = 1, 2, 3$. All three GrASP agents use a complete-tree lookahead planning procedure and use Goal-conditioned Affordances with $K = 4$ affordance mapping heads on *Fish-Swim* task. *Right:* Learning performance of GrASP agents GA-4 and A-8, model-free TD3 baseline, and GrASP agents with randomly initialized affordance modules RND GA-4 and RND A-8 on *Fish-Swim* task. The RND GA-4 and RND A-8 agents use an identical architecture to that of GA-4 and A-8, but do not learn the parameters of the affordance module.

---

**Algorithm 1** Learning Affordance Selections for Planning using GrASP

---

Initialization:

    Initialize Agent's online parameters $= \{\theta^{\text{encode}}, \theta^{\text{dynamic}}, \theta^{\text{rew}}, \theta^V, \theta^{\text{afford}}\}$ randomly

    Initialize Agent's target parameters $= \{, , , , \}$

    Synchronize target parameters with the online parameters

    Initialize Replay Buffer $D$

For each step:

    **# Agent-environment interaction loop**

    Receive an observation $x_t$ from environment

    Encode the observation to an abstract state $s_t = f^{\text{encode}}(x_t)$

    Sample an action $a_t \sim \pi(\bullet | s_t)$,

        where $\pi(\bullet | s_t)$ is the result of complete-tree or UCT search

        which uses rest of the agent modules: $f^{\text{dynamic}}, f^{\text{rew}}, f^V, f^{\text{afford}}$

    Execute $a_t$ in the environment and receive next observation $x_{t+1}$ and reward $r_{t+1}$

    Append transition $\{x_t, a_t, r_{t+1}, x_{t+1}\}$ to replay buffer $D$

    **# Sample a minibatch, use it to obtain predictions and value targets from the Agent**

    Sample a sequence of transitions of length $n$ from replay buffer

        $\{x_i, a_i, r_{i+1}, x_{i+1}, \cdots, x_{i+n}\} \sim D$

    Encode observation $x_i$ to an abstract state $s_i = f^{\text{encode}}(x_i)$

    Obtain $s_j = f^{\text{dynamic}}(s_{j-1}, a_{j-1})$, along with $f^{\text{rew}}(s_{j-1}, a_{j-1}), f^V(s_j)$,

        for $j = i, \cdots, i + n$

    Compute value target $\hat{v}_j$, for $j = i, \cdots, i + n$:

        $\hat{v}_j = r_j + \gamma r_{j+1} + \cdots + \gamma^{n-1} \max_{b \in f^{\text{afford}}(f^{\text{encode}}(x_{i+n}))} Q(f^{\text{encode}}(x_{i+n}), b)$

        where $f^{\text{encode}}, Q$ are obtained using the target parameters

    **# Update Agent's parameters**

    Minimize $\mathcal{L}^{model} = \sum_{j=i}^{i+n} \left( r_j - f^{\text{rew}}(s_j, a_j) \right)^2 + \left( \hat{v}_j - f^V(s_j) \right)^2$

        to update $\theta^{\text{encode}}, \theta^{\text{dynamic}}, \theta^{\text{rew}}, \theta^V$

    Maximize $\sum_{j=i}^{i+n} V(s_j)$ to update $\theta^{\text{afford}}$

    Synchronize target parameters with the online parameters every $T$ learning updates

---

