# OpenReview forum: "GrASP: Gradient-Based Affordance Selection for Planning"
_ICLR.cc/2022/Conference — ICLR 2022 Submitted_

### Official Review · Reviewer_VDJ5 · 2021-10-28

**Correctness:** 3
**Technical Novelty And Significance:** 3
**Empirical Novelty And Significance:** 3
**Recommendation:** 8
**Confidence:** 3

**Main Review:**

- The paper is well structured and well written.
- The proposed approach is articulated well.
- The experimental section is good.
- The authors have done a good job of explaining the challenges as the approach was developed.

- While others in the multi-agent simulation community and the RL community have looked at incorporating affordance inspired concepts into agent models there are aspects of this work which is different and novel from other previous work. The consideration of how affordances will affect the generation of the actions branches in a planning tree is important. In my mind however, the fact the paper is considering the effect of goals on the nature of affordances (what the authors call Goal-conditioned affordances) is what makes this work different.
This is important because the literature on affordance theory from ecological psychology makes it clear that affordances are goal and intention dependent. In some previous works affordances are treated as static actions, whereas the concept is dynamic and agent dependent.

- My criticisms of the paper are by no means major. I would have liked to have seen
1. a more detailed motivating example as to why incorporating affordances in this way is important.
2. the experimental domains selected could have been more sophisticated. I think the paper would have been better if there were less domains/games. Perhaps have a simple and complex domain. It is in the more complex domains (in my opinion) where it is worthwhile to incorporate a concept like an affordance.

**Summary Of The Paper:**

This paper looks incorporating the concept of affordances from ecological psychology into the planning process.
The basic premise is that affordances represent the possible relevant actions available to an agent that are potentially moderated by their goals and the state of the world. This has the potential to reduce the size of planning search trees.

In this paper the authors claim two contributions (1) the incorporation of affordance consideration into the planning process and (2) an approach for discovering affordances which the authors have called GrASP as it is a gradient based approach.

The paper implements the approach and tests it experimentally on a range of domains from the DeepMind control suite.



**Summary Of The Review:**

I think this a good paper. It builds on existing ideas on incorporating affordances but does it in a way that is novel.

---

### Official Review · Reviewer_pYew · 2021-11-03

**Correctness:** 3
**Technical Novelty And Significance:** 2
**Empirical Novelty And Significance:** 2
**Recommendation:** 5
**Confidence:** 4

**Main Review:**

Strengths:
- The paper approaches the problem of continuous control from a discrete planning approach, and shows how the discrete set of actions/options can be updated with gradient descent.

- The experiments highlight the advantage of GrASP when operating over a (separately learned) lower-dimensional option space. They also analyze the different aspects of the algorithm, including different number of heads, conditioning the affordance module on the goal, and the degree of switching between outputs.

Weaknesses:
- The proposed algorithm is only compared to TD3 in the main results. The paper acknowledges that it is not state of the art for model-based RL in continuous domains, but it would nonetheless be useful to include the comparisons to DREAMER in the main set of results, i.e., in Section 3.1 and Section 3.2. Other potential comparisons include [1] and [2].

- Overall, it seems that GrASP has the greatest advantage when operating over option spaces. This could be because the action space is already narrowed down and the planner only needs to operate over a lower-dimensional action space, e.g., the tasks in Section 3.1 have 2-D or 4-D spaces. Or perhaps, it’s due to the fact that the effective planning horizon is shortened. In Section 3.1, I’m curious how GrASP performs on higher-dimensional option spaces and on tasks with varying horizon lengths.

Finally, there are some details that are still unclear to me:
- Does the TD3 baseline in Section 3.1 also operate over the pre-trained options? It’s not clear whether TD3 is performing poorly using primitive actions or with the pre-trained options.

- In the implementation of the affordance module, is there some regularization to ensure the output actions/options do not collapse to the same output?

- I’m also curious about the discovered actions/options. Do they generally concentrate around the optimal action(s) or are they diverse among themselves? I also assume that the visualizations in Figure 2 are from the A- agents, and that the GA- agents discover something different?

[1] Chua et al. Deep Reinforcement Learning in a Handful of Trials using Probabilistic Dynamics Models. NeurIPS 2018.
[2] Bharadhwaj et al. Model-Predictive Control via Cross-Entropy and Gradient-Based Optimization. L4DC 2020.

**Summary Of The Paper:**

This paper proposes a gradient-based method for selecting affordances in planning. Specifically, it includes an affordance module that maps state representations to continuous actions/options. The planning procedure is aided by a learned model with an encoding, dynamics, reward, and value module. Most prior work uses the gradients through the trajectory under the learned model to update the policy/actions, whereas this work is about using them to update the collection of K potential actions to select from.

**Summary Of The Review:**

This paper proposes a gradient-based approach for planning in continuous control problems. However, the experiments currently lack relevant comparisons to understand the significance of the proposed approach. I’m also curious how performance varies on tasks with different option spaces and different horizons.

---

### Official Review · Reviewer_ohYy · 2021-11-03

**Correctness:** 3
**Technical Novelty And Significance:** 3
**Empirical Novelty And Significance:** 2
**Recommendation:** 5
**Confidence:** 2

**Main Review:**

# Strengths

- Detailed ablation analysis of the proposed approach with shallow-depth complete trees and UCT-based MCTS.
- Extended the Value equivalent model with affordance learning module
- Qualitative analysis on the switches between affordance mapping outputs. Take-away: across the GA-k (k > 1) episode returns are positively skewed when those agents
outperform GA-1 and TD-3.

# Weakness

- The claim that the paper introduced "the idea of discovering affordances for planning in continuous domains" seems inconsistent. The paper discusses Sampled MuZero in related work, that extends MuZero to continuous action space. In contrast, GrASP uses gradients through the planning process to discover a discrete set of affordance mappings which are used for building the tree. How is the discrete set of affordance decided from continuous action and state space? It seems like a design choice specific to each environment.
- The paper does not compare the proposed approach with other SOTA model-based algorithms. While the authors mention "GrASP algorithm is not the SOTA model-based RL in continuous domains", it is still difficult to say how significant the contribution of the proposed approach is.
    - GrASP is compared to another model-based algorithm Dreamer in the Appendix. This seems like a relevant comparison, however some observations like "in *Ball-In-Cup-Catch* it seems to be learning relatively faster than Dreamer" seem inconsistent with the plots shown. Dreamer performs better than GrASP on Walker:walk and cheetah:run. A discussion to analyze this relative performance between GrASP and Dreamer will be helpful to understand the merits of each of the models.
    - The result section highlights which of the different variants of GrASP perform best in different scenarios. While this looks like a detailed ablation analysis, it does not provide the reviewer/reader with
- Action and option affordance learning ablation is compared to a just model-free baseline (TD3). There seems no take-away message on what model works the best, and is task environment/hyperparameter dependent.
- The claim "*Cartpole-Swingup* is the only domain where all configurations of GrASP fail to improve over the TD3 baseline agent." seems incomplete. While at least one of the GrASP variants learn faster than TD3, the asymptotic performance for TD3 seems higher in reacher:easy and walker:walk.

# Minor

- Pg 5 "Each trajectory starts at the root ~~note~~ node."
- [Clarification] How to compare a discrete time "step" in TD3 vs GrASP?

**Summary Of The Paper:**

The paper proposes a gradient-based affordance selection, i.e., action/option selection, in addition to the value equivalent modules for tree-expansion procedure(s) in planning.
The claim is that GrASP can learn primitive-action and option selection and plan in a continuous state and action space to outperform model-free RL.
The affordance module maps the state to K actions or options from continuous space. These affordances are later used for expanding the look-ahead search tree. The key idea is to compute the gradient of performance loss with respect to the affordance model's parameters through the planner. The method uses two expansion procedures: (1) shallow-depth complete trees, (2) UCT-based MCTS. The performance of TD3 is compared to different variants of the proposed algorithm on some tasks from the DM Control Suite.

**Summary Of The Review:**

While I enjoyed the addition of affordance module in value equivalent network and found it interesting contribution to integrate learning and planning, the experiments do not analyze all the merits of the proposed approach as compared to prior work. It is unclear how design choices like the discrete set of actions/options is constructed from the continuous space.

---

### Official Review · Reviewer_LdeK · 2021-11-03

**Correctness:** 3
**Technical Novelty And Significance:** 3
**Empirical Novelty And Significance:** 3
**Recommendation:** 6
**Confidence:** 3

**Main Review:**


Strength:
- As far as I am aware, this is a novel contribution and an interesting idea overall. Learning affordances seems to be closely related with learning options, which is a must for an agent to be able to learn how to perform complex tasks.
- The paper is clearly written and is easy to follow. A reader with some moderate knowledge of RL should be able to re-implement the methods in this work.

Weakness:
- One thing is not quite clear to me:  what prevents the K heads from outputting similar actions? It could be possible that the initial set of affordances predicted by the model are quite similar, and it gets trapped in some local optimum that prevents the model from trying out other options. Does initialization play a big role?
I feel like either I am missing something from the paper, or there is a little bit more context that's needed to understand when this method would work or fail.
I would appreciate if the authors could clarify this.

- The gradient update on affordance module. The summation over b is only considering K affordances, but if the total number of possible affordances is larger than K and you constraint to only K, you might never explore any of the one previously left out. How do you deal with this? Is there some type of exploration that's needed here?

- Figure 4 is hard to interpret. I understand it is trying to show the selection across different affordances but maybe a line plot like this is not the right representation. Also, what am I supposed to get from this plot? Is switching often good? Is it bad? Some clarification around this plot would be useful.

**Summary Of The Paper:**

The paper proposes dealing with tree search planning in continuous action spaces by learning affordances.
The authors propose an architecture composed of several modules, where they are able to plan ahead in a tree search manner by learning a module that expands K discrete possible affordance from each state node.
The results suggest that the proposed technique is able to deal with difficult tasks in which standard tree search methods would not be easy to apply.

**Summary Of The Review:**

Overall an enjoyable read with interesting ideas. There might be some context or clarification missing, I am not convinced that the objective propose would explore available affordances/actions, and would not get stuck looking at a suboptimal subset.
If that is the case, it would be useful to know what are the requirements to prevent this from happening.

---

### Decision · Program_Chairs · 2022-01-20

**Decision:**

Reject

**Comment:**

The paper proposes a learning framework that allows for planning in continuous action spaces using tree search. Key to the approach is performing tree search over a discrete set of learned affordances that provide a compact abstraction of the action space that facilitates planning. The affordances are learned by passing gradients through a model-based planner that uses learned models of the dynamics, reward, and state-value functions. Experimental evaluations demonstrate the ability to perform tree search-based planning using the learned affordances in a variety of domains for which tree search would otherwise be difficult.

The paper is topical, both with regards to its consideration of affordances as temporal abstractions that facilitate planning as well as the broader notion of integrating planning and learning. Several reviewers agree that the means by which affordances are learned by passing gradients through the planner is both interesting and novel. The reviewers also emphasize that the paper is well written and easy to follow, and that the approach is reproducible as a result. The reviewers raised a few concerns with the initial submission, notably the need for experimental comparisons to other recent baselines, which are important to clarifying the significance of the contributions, and the susceptibility to collapse in the affordance distribution. The authors clarified some of these questions and proposed adding comparisons to other baselines (e.g., DREAMER, for which there is already a comparison in the appendix), however it is not clear whether the submission was updated accordingly. The authors are encouraged to take this feedback into account and to include a more thorough experimental evaluation in any future version of the paper.